# Feasibility of Combining Disease-Specific and Balance-Related Measures as Risk Predictors of Future Falls in Patients with Parkinson’s Disease

**DOI:** 10.3390/jcm12010127

**Published:** 2022-12-23

**Authors:** Chang-Lin Tsai, Yun-Ru Lai, Chia-Yi Lien, Chih-Cheng Huang, Wen-Chan Chiu, Yueh-Sheng Chen, Chiun-Chieh Yu, Ben-Chung Cheng, Yi-Fang Chiang, Hsueh-Wen Chang, Cheng-Hsien Lu

**Affiliations:** 1Department of Neurology, Kaohsiung Chang Gung Memorial Hospital, Chang Gung University College of Medicine, Kaohsiung 83301, Taiwan; 2Department of Hyperbaric Oxygen Therapy Center, Kaohsiung Chang Gung Memorial Hospital, Chang Gung University College of Medicine, Kaohsiung 83301, Taiwan; 3Department of Internal Medicine, Kaohsiung Chang Gung Memorial Hospital, Chang Gung University College of Medicine, Kaohsiung 83301, Taiwan; 4Department of Radiology, Kaohsiung Chang Gung Memorial Hospital, Chang Gung University College of Medicine, Kaohsiung 83301, Taiwan; 5Department of Biological Science, National Sun Yat-sen University, Kaohsiung 80424, Taiwan; 6Department of Neurology, Xiamen Chang Gung Memorial Hospital, Xiamen 361126, China

**Keywords:** center of pressure, future falls, levodopa equivalent daily dose, postural sway, Parkinson’s disease, Tinetti balance and gait score

## Abstract

Evidence supports the view that postural sway in a quiet stance increases with clinical disease severity and dopaminergic therapy in idiopathic Parkinson’s disease (PD), which, in turn, increases the risk of falling. This study evaluated the feasibility of combining disease-specific and balance-related measures as risk predictors for future falls in patients with PD. The patients with PD underwent postural sway measurements (area, length, and velocity traveled by the excursion of the center of pressure) and clinical functional scores (Parkinson’s Disease Rating Scale [UPDRS] and Tinetti balance and gait score assessment) in both the on- and off-states of dopaminergic therapy. The outcome was defined as the development of a new fall. The sway area, velocity, and length increased after the medication administration. The Cox proportional hazards model showed that only previous fall history, Tinetti balance and gait score (on-state), and levodopa equivalent daily dose (LEDD) were associated with the development of future falls. The cumulative risk of fall development showed that the sway length and velocity were associated with future falls after more than six months. The combined LEDD, Tinetti balance and gait score (on-state), and velocity and length of postural sway (on-state) had the highest diagnostic accuracy (area under the curve = 0.9, *p* < 0.0001). Dopaminergic therapy can improve clinical functional scores but worsen balance-related measures. Increased sway length and velocity during the medication state are hallmarks of future falls, particularly in advanced PD. Combining disease-specific and balance-related measures can serve as an auxiliary diagnosis as risk predictors for future falls.

## 1. Introduction

The maintenance of balance in the human body requires three distinct processes: sensory organization composed of proprioceptive, visual, and vestibular systems integrated within the central nervous system; motor adjustment to execute coordinated and properly scaled neuromuscular responses; and background muscle tone [1]. Both sensory integration and motor output impairment in patients with idiopathic Parkinson’s disease (PD) contribute to postural instability [1]. These axial features cause significant disability resulting from falls [2], with the severity increasing with the disease progression. Therefore, investigating the risk factors for falls and potential strategies for their prevention is of great importance.

Human balance can be maintained by the mechanics of the body’s effects and coexists with the processes in the central nervous system engaged to solve [3]. This implies that the neural control mechanisms corresponding to the mechanics of the stance can generate control commands to stabilize the body, preventing falls. Postural control impairment reduces the stability limits. The measurement of postural sway can be used to quantify impairments in postural control and balance. Regarding the mechanics of the stance, studies on quantified postural stability during a quiet stance in subjects with PD have often yielded contradictory and limited results [4,5,6,7,8]. These discrepancies across various studies may be explained by differences in the research design (e.g., different disease severity, patients, and controls, and with or without levodopa therapy). The variables evaluated during an excursion of the center of pressure (COP) in a quiet stance were the most reported biomechanical measures. The parameters demonstrating a relationship between postural sway and PD include the area (cm^2^) [1,9], length, velocity, and variability of the velocity in the anterior–posterior (AP) COP excursions [5], the direction (AP and mediolateral (ML) [6,7,8,10]) of the excursion (mm) of the COP, and the variability of the length traveled by the COP. In addition to the parameters of quantified postural stability, studies have shown that postural sway is significantly greater in patients with severe or prolonged disease progression [2,7,9,10]

A clinical study demonstrated that both higher Parkinson’s Disease Rating Scale (UPDRS) total scores and increased sway areas were significantly associated with the risk factors for recent falls in patients with PD [9]. In addition to sensory integration and motor output impairment, the effect of dopaminergic therapy on static balance and gait remains controversial [11,12]. Dopaminergic therapy, including carbidopa and dopamine agonists, can improve postural instability in the early stages of PD; however, their effectiveness decreases with disease progression and in the presence of levodopa-induced dyskinesia [13]. Furthermore, a double-edged sword has been demonstrated for balance and gait, as increases in postural sway after levodopa dosing have been observed [8,14]. The effects of dopamine on balance and gait involve separate human supraspinal locomotor and postural control networks, rather than being considered as one function [15]. The clinical functional scores also lack sufficient items (e.g., UPDRS and Tinetti balance and gait total scores) to evaluate standing postural sway as balance measures [16]. Thus, these findings highlight the idea that combining disease-specific (e.g., clinical functional score) and balance-related measures (e.g., postural sway) as risk predictors of postural instability for future falls in patients with PD could be more satisfactory.

According to our hypothesis, clinical disease severity is proportional to sensory integration, motor output impairment, and the levodopa equivalent daily dose (LEDD) [17]. Dopaminergic therapy increases the severity of postural sway, especially in patients with advanced PD. We tested the feasibility of combining disease-specific and balance-related measures as risk predictors to prevent future falls.

## 2. Patients and Methods

### 2.1. Study Design and Patient Selection

The participants met the International Parkinson and Movement Disorder Society clinical diagnostic criteria for idiopathic PD, as well as the magnetic resonance imaging criteria [18,19]. The patients enrolled in this study included those who had been followed up for more than six months after titration of their daily anti-Parkinsonian agents to a steady dose, and whose Hoehn and Yahr stages were between 1 and 3 and could walk independently [20,21]. The patients with severe disability or needing a wheelchair unless aided (Hoehn and Yahr staging ≥ 4), a cognitive decline that precluded following our instructions (Clinical Dementia Rating of more than or equal to 1), and any etiologies that were severe enough to interfere with balance (e.g., visual, vestibular, and proprioceptive problems and lower limb weakness) were excluded. This study was approved by the Institutional Review Committee on Human Research of the hospital (IRB 201901802B0). All the participants understood the purpose of this study and signed an informed consent form. A total of 145 patients were prospectively evaluated. Finally, 95 patients were included in the analysis (Figure 1).

### 2.2. Clinical Assessment of PD

Clinical assessments, including the clinical rating scale assessments and postural sway measures, were performed during both the off- and on-states of medication. The off-state of medication was defined as 12 h after the latest dose of anti-parkinsonism agents, and the on-state as at least one hour after taking the anti-parkinsonism agents. A complete history was recorded for all the patients, including their age at disease onset, sex, body mass index (BMI), disease duration, LEDD [17], and anti-Parkinsonian medications. The clinical severity of PD was assessed using the Hoehn and Yahr stages [20,21], the Unified Parkinson’s Disease Rating Scale (UPDRS) (UPDRS subscores I, II, III, IV, and UPDRS-derived postural instability and gait difficulty (PIGD) score) [22], and the Cognitive Abilities Screening Instrument (CASI C-2.0) [23]. The presence or absence of dyskinesia was based on the UPDRS IV (motor complications). The Tinetti balance and gait scores were used to assess the risk of falling in patients with PD [16]. This consists of two parts, the balance score (score = 0–16) and gait score (score = 0–12), and ranges from 0 to 28 points by summing the points obtained for each of the two tests.

### 2.3. Measurement of Postural Sway

The postural sway measurements were performed using a TekScan MatScan pressure mat model 3150 (TekScan Inc., South Boston, MA, USA) in conjunction with the Sway Analysis Module (SAM) software used to analyze the sway data (Figure 2). Following the device calibration, the measurement of each patient was recorded three times during normal standing with eyes open in the off- and on-medication states, without any prior practice sessions, under standardized conditions. In total, there were six tests for each patient [off-state (*n* = 3) and on-state (*n* = 3)]. The duration of each recording was 30 s for each test. The mean values of three successive recordings were averaged and used in the analyses. During the measurements, the patients stood with no shoes on, with their feet together, and arms by their sides. The patients were asked to look straight ahead at a mark on the wall facing them. The sway parameters recorded by the mat were area (cm^2^), length (cm), velocity (cm/s), length y, and length x (length in anteroposterior (AP) and mediolateral (ML) directions) traveled by the excursion of the center of pressure (COP).

### 2.4. Assessment of the Moment of Inertia

The moment of inertia (I) is the name given to rotational inertia, the rotational analog of mass for linear motion. It appears in the relationships of the dynamics of rotational motion. The value of the inertia (I) for each subject was calculated by mh^2^, where m and h were the mass (body weight) and height of the subject. It was standardized as a z-score in each subject, respectively, prior to the correlation analysis [24]. As it is well-known that the angular momentum (L) of a body is always conserved, the moment of inertia (I) and angular velocity (ω) are inversely proportional to each other. (L = I × ω).

### 2.5. Falls Assessments

In this study, previous falls were defined as having had at least one fall, either indoors or outdoors, during the preceding 12 months before enrollment [25], and the severity, frequency, and circumstances of falling were recorded [26]. All the patients were enrolled at baseline and at regular follow-ups at the neurology outpatient clinic every three months. The outcome was defined as the presence of new falls after at least six months of follow-up.

### 2.6. Statistical Analysis

The data are expressed as the mean ± SD or median (interquartile range [IQR]). The continuous variables that were not normally distributed were logarithmically transformed to improve normality before the analysis. The categorical variables were compared using the chi-square or Fisher’s exact tests. Levodopa-induced dyskinesia is a well-known risk factor for increased postural sway [10,27]. To compare the effect of levodopa-induced dyskinesias on the postural sway measurements, the patients were divided into two groups, based on the presence or absence of dyskinesias. The parameters of postural sway among the patients with and without dyskinesia and the healthy controls were compared using a one-way ANOVA. The changes in the parameters of postural sway and clinical scores between the two groups (patients with or without dyskinesia) before and after dopaminergic therapy were compared using repeated-measure ANOVA. The risk factors for future falls that occurred after a minimum follow-up period of six months were determined using Cox’s proportional hazards model. The association between the high and low sway length and velocity, divided by the median value (≥median and <median value), and the survival curve between the presence of future falls were assessed using Kaplan–Meier plots and compared using the log-rank test. The correlation analyses were used to test the influence of the moment of inertia on the parameters of standing postural sway during the off- and on-states of medication. Since the moment of inertia can affect the sway velocity, a partial correlation analysis was used to determine the relationship between the length and velocity of postural sway and LEDD, the UPDRS total scores, and the Tinetti balance and gait scores, after controlling with the moment of inertia. The area under the ROC curve (AUC) for the presence of prospective falls, in addition to the sensitivity, specificity, and Youden’s index of each parameter, was also calculated. Finally, the statistical significance of the difference between the two AUCs (each component and the combination derived from the same cases) was evaluated using the method described by DeLong et al. [28]. All the statistical analyses were conducted using the IBM SPSS Statistics v23 statistical software (IBM, Redmond, WA, USA).

## 3. Results

### 3.1. General Characteristics of Patients with PD

This study evaluated 95 patients with PD, including 48 women (mean age, 66.2 ± 9.7 years) and 47 men (mean age, 69.6 ± 9.7 years). A previous fall history was present in 38 cases (40%). The mean disease duration was 5.5 ± 4.9 years, and the LEDD (mg) was 815.6 ± 575.3 mg. The mean UPDRS I, II, III, and IV subscores, PIGD scores, and total scores were 1.7 ± 1.4, 10.0 ± 5.3, 17.8 ± 9.5, 9.3 ± 5.4, 4.7 ± 3.5, and 38.7 ± 19.1, respectively. The severity, frequency, and circumstances of falling in those patients who had a previous fall history are listed in Table 1.

### 3.2. L-Dopa-Induced Dyskinesia Increases Postural Sway

The standing postural sway between patients with PD with and without dyskinesia before and after dopaminergic therapy is shown in Table 2. The patients with dyskinesia had a longer disease duration and LEDD than those without dyskinesia (*p* < 0.0001 and *p* < 0.0001, respectively). Regarding the postural sway parameters among the three groups (PD without dyskinesia, PD with dyskinesia, and healthy controls), the PD with and without dyskinesia during the off-state had a higher sway area, velocity, length, and length (x) than those of the healthy controls (*p* = 0.007, *p* = 0.049, *p* = 0.048, *p* = 0.0001, one-way ANOVA). The sway area, velocity, length, length (y), and length (x) did not differ significantly between the PD patients with and without dyskinesia during the off-state (*p* = 0.33, *p* = 0.9, *p* = 0.94, *p* = 0.44, and *p* = 0.49, respectively). However, the PD patients with dyskinesia showed a significantly higher sway area, velocity, length, length (y), and length (x) than those with PD without dyskinesia after dopaminergic therapy (*p* = 0.001, *p* = 0.004, *p* = 0.002, *p* = 0.01, and *p* = 0.01, respectively). Furthermore, the parameters of the postural sway, including area, length (x), and length (y) before and after dopaminergic therapy, between the patients with and without dyskinesia showed statistical significance (*p* = 0.006, *p* = 0.024, and *p* = 0.013, respectively), while the differences in length and velocity in the posture sway were not significant (*p* = 0.086, and *p* = 0.11).

### 3.3. Dopaminergic Therapy Improved Functional and Fall Risk Scores

The UPDRS-derived PIGD score showed statistical significance during the off- and on-states between the patients with and without dyskinesia (*p* < 0.0001 and *p* = 0.03, respectively). Regarding the clinical severity and fall risk scores, the UPDRS III, PIGD, total score, Tinetti balance, gait, and total scores in PD with dyskinesia showed a significant improvement after dopaminergic therapy (*p* < 0.0001, *p* < 0.0001, *p* < 0.0001, *p* < 0.0001, *p* = 0.002, and *p* < 0.0001, respectively). The UPDRS III, PIGD, total score, Tinetti balance, gait, and total scores in the patients with PD without dyskinesia also showed significant improvement after dopaminergic therapy (all *p* < 0.0001). However, only the UPDRS-derived PIGD and total scores before and after dopaminergic therapy between the patients with and without dyskinesia showed statistically significant differences (*p* = 0.002 and *p* = 0.006, respectively), while the UPDRS III and Tinetti balance, gait, and total scores showed no significant difference (*p* = 0.47, *p* = 0.82, *p* = 0.40, and *p* = 0.58, respectively).

### 3.4. Risk Factors Associated with Subsequent Falls in the Cox Proportional Hazards Model

The follow-up duration in 63 cases was more than 12 months, while it was less in the other 32. The mean follow-up duration in our patients was 12.4 ± 2.8 months. We compared the baseline clinical features, functional scores, and parameters of standing postural sway during the on- and off-states in the patients with PD, with and without future falls, after >6 months of follow-up using the univariate Cox proportional hazards model (Table 3). The significant univariates in the univariate Cox proportional hazards model were the disease duration (years) (*p* = 0.001), LEDD (*p* < 0.0001), UPDRS total score during the on- and off-states (*p* < 0.0001 and *p* < 0.0001, respectively), Tinetti balance and gait score during the on- and off-states (*p* < 0.0001 and *p* < 0.0001, respectively), and length and velocity of postural sway (on-state) (*p* = 0.011 and *p* = 0.016, respectively). When these significant variables were applied in the Cox proportional hazards model, only the previous fall history (*p* < 0.0001, 139.4 (3.6-HR: 139.4, 95% CI: 3.6–5392.0), LEDD (*p* = 0.008, HR:1.1, 95% CI:1.0–1.01) and Tinetti balance and gait score during the on-state (*p* = 0.04, HR:0.82, 95% CI:0.74–0.91) were independently associated with prospective falls. The velocity and length of postural sway during the on-state were classified into high- and low-velocity and high- and low-length, respectively, according to the median value in our cohort, and the cumulative risk of development of future falls was assessed over six months using the Kaplan–Meier method. The results showed that a high- or low-velocity and high- and low-length of postural sway could predict the development of prospective falls over six months (*p* = 0.013 and *p* = 0.023, log-rank test) (Figure 3A,B).

### 3.5. Partial Correlation Analyses of Standing Postural Sway, LEDD, and UPDRS Total Score after Controlling with Moment of Inertia

Our results showed that both higher Tinetti balance and gait scores and UPDRS total scores, longer PD duration, higher LEDD dosage, and both increased sway length and velocity were significantly associated with future falls in patients with PD (Table 3). The correlation analyses were used to test the influence of the moment of inertia on the parameters of standing postural sway during the off- and on-states of medication, and the statistical analysis (r, *p*-value) was as follows: area (cm^2^) (r = −0.05, *p* = 0.62), velocity (cm/s) (r = −0.10, *p* = 0.34), length (cm) (r = −0.10, *p* = 0.34), length y (cm) (r = −0.05, *p* = 0.65) and length x (cm) (r = −0.09, *p* = 0.39) of standing postural sway during off-state of medication, and area (cm^2^) (r = −0.07, *p* = 0.5), velocity (cm/s) (r = −0.16, *p* = 0.11), length (cm) (r = −0.16, *p* = 0.13), length y (cm) (r = −0.06, *p* = 0.54)and length x (cm) (r = −0.16, *p* = 0.13) of standing postural sway during on-state of medication. Further, the partial correlation analyses used to test the influence of the parameters of standing postural sway during the on-state and LEDD, UPDRS total score, and Tinetti balance and gait score, after controlling with the moment of inertia, are listed in Table 4. The length (cm) and velocity (cm/s) during the on-state were significantly correlated with the LEDD, UPDRS total scores, and Tinetti balance and gait scores (*p* < 0.0001 and *p* < 0.0001, *p* < 0.0001 and *p* < 0.0001, and *p* < 0.0001 and *p* < 0.0001, respectively). Furthermore, the LEDD was significantly correlated with the disease duration, UPDRS total scores, and Tinetti balance and gait scores (*p* < 0.0001, *p* < 0.0001, and *p* = 0.02, respectively).

### 3.6. Diagnostic Accuracy for Predicting the Presence of Prospective Falls, Using Receiver-Operating Characteristic Curve Analysis

We tested our hypothesis that dopaminergic therapy increases the severity of postural sway, especially in patients with advanced PD, and the feasibility of combining disease-specific and balance-related measures as risk predictors to prevent future falls. In this study, we only selected the significant variables from the Cox regression model, including the LEDD and Tinetti balance and gait score (on-state), and significant univariate postural sway from the univariate Cox regression model, including the velocity and length in a postural way (on-state) (Table 3) as candidate variables, to assess the diagnostic accuracy of falls in the ROC analysis (Table 5). We further evaluated these variables in univariate logistic regression for the risk of future falls, and the statistical analysis was as follows: LEDD (*p* < 0.0001), Tinetti balance and gait score (on-state) (*p* = 0.001), and velocity, and length in a postural way (on-state) (*p* = 0.005 and *p* = 0.004). Finally, statistical analyses for the LEDD, Tinetti balance and gait score (on-state), velocity, and length in a postural way (on-state), and their combination in predicting the prospective falls using a ROC analysis, were conducted, and the results were as follows: LEDD, *p* < 0.0001; Tinetti balance and gait score (on-state), *p* < 0.0001; velocity and length in the postural sway (on-state) (*p* = 0.001 and *p* = 0.001, respectively, and their combination, *p* < 0.0001). The sensitivity, specificity, and AUC using the ROC curve analysis are presented in Table 5. Pairwise comparisons of ROC curves of LEDD, Tinetti balance and gait score (on-state), velocity and length in a postural manner (on-state), and their combination, for prospective falls were performed using the DeLong method, as shown in the footnote of Table 5. The AUC of the combined LEDD, Tinetti balance and gait score (on-state), and velocity and length in postural sway (on-state) were higher than those of the LEDD, Tinetti balance and gait score (on-state), and velocity and length in postural sway (on-state) alone (*p* = 0.047, *p* = 0.02, *p* = 0.005, and *p* = 0.006, respectively). The sensitivity and specificity of this combination were higher than those of LEDD alone (sensitivity,91.3% vs. 82.6%; specificity,80% vs.74.3%) (Figure 4).

## 4. Discussion

### 4.1. Major Findings

The present study showed that dopaminergic therapy can improve clinical functional scores but worsen balance-related measures. Further, the study confirmed the hypothesis that combining disease-specific (Tinetti balance and gait score and LEDD) and balance-related measures (length and velocity in postural sway) can serve as an auxiliary diagnosis of future falls in terms of the sensitivity, specificity, and AUC using a ROC analysis [29]. Additionally, it highlights the importance of both the underlying PD severity and dosage of dopaminergic drugs on postural sway and risk of falls. The LEDD is not only a surrogate marker reflecting disease severity in many cases [17], but also reflects the increased severity of postural sway in patients with PD.

Several clinical balance scales and objective balance tests and /or their combinations have been used to assess postural stability and the risk of falling in patients with PD [9,15,30,31,32]. The first study was to determine the efficacy of clinical static and dynamic posturography and balance scales in detecting postural instability and discriminating between fallers and non-fallers in PD patients in the on-state [30]. This study showed that this combination of clinical and posturographic measures, especially dynamic posturography, in their best on-state would be useful in the prospective assessment of fall risk in PD patients. The second study collected the risk factor, and clinical disease severity (e.g., UPDRS score) and tested for static balance using an inclinometric device including eyes-open and eyes-closed in the on-state [9]. This study showed that a higher UPDRS total score and an increased postural sway area were independent risk factors for recent falling in PD. The third study enrolled people with early-stage PD and undertook a battery of neurologic and functional tests in their optimally medicated state [16]. This study showed the UPDRS total score, total FOG score, occurrence of symptomatic postural orthostasis, Tinetti total score, and length of postural sway in the anterior–posterior direction were significant factors for predicting prospective falls [16]. The fourth study aimed to explore the interaction between dopaminergic replacement medication, and the focus of attention on postural stability in patients with early stage PD and showed that an absence of dopamine exacerbates dysfunction. Besides the combination of both clinical scales and objective balance tests, the three simple clinical tests [31] and self-administered questionnaires which targeted the fear of falling were also validated to assess the prospective risk of falling in patients with PD [32]. These discrepancies across previous studies [9,16,30,31,32] and ours may be explained by differences in the research design (e.g., different disease severity, different study groups with or without a control, with and/or without levodopa therapy, different clinical scales, different objective balance tests, and follow-up periods).

### 4.2. The Effects of Clinical Disease Severity on Standing Postural Sway

The evidence suggests that clinical disease severity plays a critical role in postural instability, which increases the risk of falling [9,13,33]. Regarding the parameters traveled by the COP during a quiet stance, one study showed that the length x (cm) displacement was abnormal in the early stage of PD, which may be a compensatory mechanism for postural control before clinical evidence of postural instability [33]. Medication increased the length x (cm) of postural sway [6,33], and it was also positively correlated with both the PD duration and clinical disease severity [6]. Another study showed that both the clinical disease severity and sway area during the on-state were independent risk factors associated with recent falls in PD [9]. Our study showed that higher lengths (cm) and velocity (cm/s) of postural sway during the on-state could predict future falls (*p* = 0.023 and *p* = 0.013, log-rank test).

### 4.3. The Effects of Dopaminergic Therapy on Standing Postural Sway

Although levodopa improves postural stability, treatment with levodopa increases postural sway abnormalities in patients with PD, particularly under prolonged PD duration and more severe PD functional outcomes [34]. Dopaminergic therapy has been shown to reduce tonic firing rates in the globus pallidus internus (GPi) [35,36], reflected in downstream targets, and improve the motor adjustment component of postural control [37]. However, it may cause further bursting and irregular firing patterns in the basal ganglia after medication, and dopaminergic medication-induced dyskinesia may be partially related to the same pathophysiological process [38]. These pathophysiological changes in the basal ganglia may be detrimental to the sensory organizational control of posture [37]. Levodopa-induced dyskinesia can occur in advanced PD and may lead to more swings during the on-state, compromise balance, and contribute to postural instability and falls. Our study demonstrated that all the parameters of postural sway traveled by the COP in both the dyskinesia and non-dyskinesia groups in the quiet stance increased after the dopaminergic therapy. In contrast to other studies [33], only those parameters during the on-state were predictive of falls, while those during the off-state in our study were not. Clinical studies have shown that surgery can not only reduce firing rates, thereby releasing downstream nuclei from inhibition and improving motor function, but also normalize firing patterns in the basal ganglia [38]. Deep brain stimulation (DBS) alleviates the cardinal symptoms, including tremor, rigidity, and bradykinesia, but it does not offer the same improvement on the symptoms of PIGD. This may imply that the axial and distal controls are differentially affected by the DBS. The effects of DBS on both the subthalamic nucleus (STN) and globus pallidus interna (GPi) can continue over five years during both the off- and on-medication states. Although DBS can improve PIGD initially, its effect declines over time. Regarding the differences in long-term effects between STN and GPi DBS, the GPi DBS, in combination with levodopa, seemed to preserve PIGD better than the STN DBS [39,40]. The selection of DBS candidates is a complex task. Whether symptoms are L-dopa responsive or not, how well they respond, the axial symptoms vs. the rigidity/bradykinesia, and whether there are comorbid cognitive psychiatric conditions need to be taken into consideration.

### 4.4. Strengths and Limitations

This study had two strengths. First, our results highlight the feasibility of combining disease-specific (Tinetti balance and gait score and LEDD) and balance-related measures (length and velocity in postural sway) to serve as an auxiliary diagnosis of future falls. Second, although dopaminergic therapy may improve postural instability, it can increase postural sway, especially in advanced-stage PD (e.g., higher LEDD and UPDRS scores, and the presence of drug-induced dyskinesia). This highlights the limitations of current dopaminergic therapies and the need to improve interventions that target postural instability.

Furthermore, our study had four limitations. First, the follow-up duration was short, and it is better to track disease progression using both quantitative biomechanical measures and clinical scores to predict the long-term outcome of future falls. Second, patients with advanced PD or mild-to-moderate dementia were excluded. Thus, there is some uncertainty in assessing the role of postural sway in non-selected patients with PD and falls. Third, although clinical scores have been the most widely used rating scale for tracking PD severity and balance and gait assessment, the scoring of the clinical scores could be subjective and affected by the examiner’s experience. The silent postural sway velocity and sway length were objective balance-related measures and were statistically significant in the univariate Cox hazard model, but they were not statistically significant in the multivariate analysis of the Cox hazard model. Fourth, the time course of the COP motion during quiet standing, corresponding to body sway, is of critical importance for understanding the neural mechanisms of postural control [23]. A posturographic analysis with descriptors that can be considered universal, thus related to the effective changes in motor control strategies of the patients rather than to the mechanics of the stance, is needed. Finally, regarding the falling situations, walking is the most frequent (25/38, 65.8%), followed by sitting/standing (11/38, 26.3%), and turning (3/38, 7.9%). Although increased postural sway during a quiet stance is a risk factor for prospective falls [41] and a marker of progression in a longitudinal study [10], most of the falls in our patients with PD occur during a walking rather than a static state and, thus, the dynamic balance might be more reasonable to predict future falls in PD patients than the static balance.

## 5. Conclusions

Dopaminergic therapy is a double-edged sword for improving balance and gait. Dopaminergic therapy can improve clinical functional scores but worsens balance-related measures. Our results highlight the feasibility of combining disease-specific and balance-related measures that can serve as an auxiliary diagnosis as risk predictors of future falls in patients with PD. Monitoring the changes in the quantitative postural sway measurements to track disease progression during longitudinal follow-up can provide a scientific basis for dopaminergic therapy, balance training programs, and the selection of high-risk candidates for surgical treatment to reduce future fall risk.

## Figures and Tables

**Figure 1 jcm-12-00127-f001:**
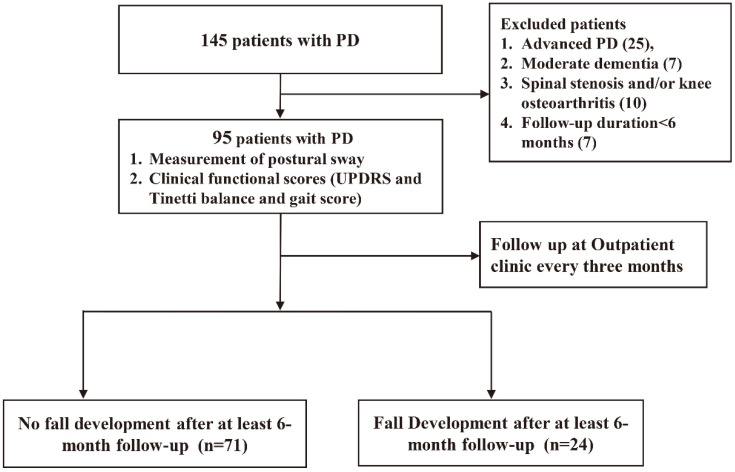
Enrollment of patients.

**Figure 2 jcm-12-00127-f002:**
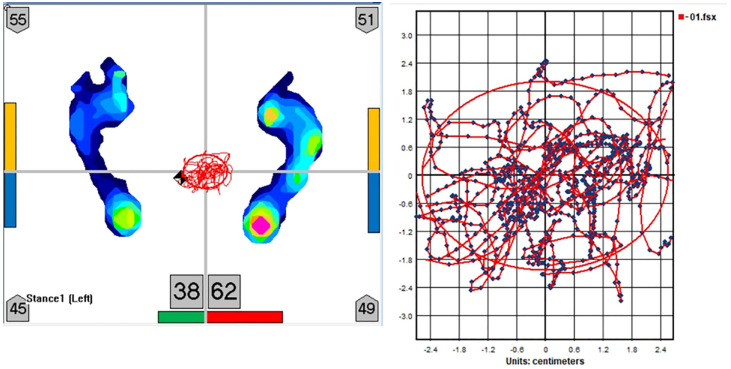
Center of pressure trajectories in the horizontal plane of a representative subject with Parkinson’s disease.

**Figure 3 jcm-12-00127-f003:**
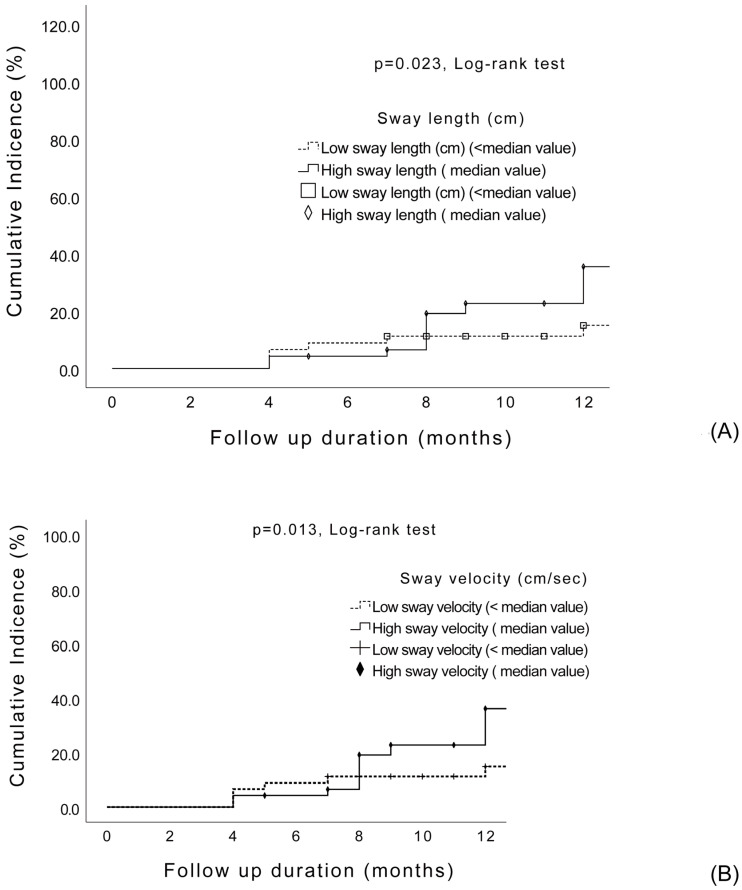
The cumulative risk of sway length (**A**) and velocity (**B**) for the development of future falls in patients with PD.

**Figure 4 jcm-12-00127-f004:**
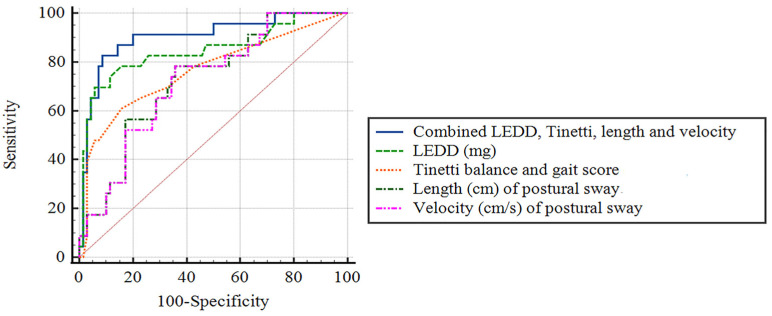
Receiver-operator characteristic curves for predicting the presence of prospective falls. The diagnostic accuracy of the LEDD, UPDRS total score (on-state), and length and velocity in standing postural sway (on-state), and its combinations, is shown based on the receiver operating characteristic curve analysis.

**Table 1 jcm-12-00127-t001:** Baseline characteristics of patients.

	Previous Falls		*p*-Value
	Non-Fallers (*n* = 57) (*n*, %)	Fallers (*n* = 38) (*n*, %)	
Age, years	67.6 ± 10.1	69.0 ± 9.5	0.48
Sex (men/women)	32/27	15/21	0.16
Body mass index (kg/m^2^)	25.4 ± 4.2	24.9 ± 4.1	0.62
Disease duration, years	4.0 ± 3.5	7.7 ± 5.0	<0.0001
Presence of dyskinesia	8	12	0.03
Freezing of gait	14	22	0.001
Orthostatic hypotension	5	1	0.4
Fear of fall	14	25	<0.0001
UPDRS total score (off-state)	34.2 ± 18.8	46.4 ± 16.9	0.002
UPDRS I (Mentation, behavior, and mood)	1.4 ± 1.0	2.2 ± 1.7	0.01
UPDRS II (ADL score)	8.0 ± 4.5	13.0 ± 5.1	<0.0001
UPDRS III (motor score) (off-state)	16.7 ± 10.3	19.9 ± 8.0	0.11
UPDRS IV (motor complications)	8.1 ± 5.3	11.3 ± 5.1	0.004
UPDRS-derived PIGD score	3.4 ± 2.6	7.0 ± 3.6	<0.0001
Hoehn and Yahr Staging	1.9 ± 0.8	2.4 ± 1.0	0.006
Cognitive Abilities Screening Instrument	83.5 ± 14.3	81.7 ± 12.4	0.59
Tinetti balance score	9.5 ± 2.8	6.9 ± 2.9	<0.0001
Tinetti gait score	14.4 ± 2.6	12.2 ± 3.4	0.001
Tinetti balance and gait score	24.0 ± 5.0	19.1 ± 2.6	<0.0001
Anti-Parkinsonian medications ^Φ^			
LEDD (mg/day)	604.3 ± 365.4	1146.9 ± 684.2	<0.0001
Levodopa	50 (87.7%)	24 (63.2%)	
Dopamine agonist (Pramipexole/Ropinirole)	35 (61.4%)	28 (73.7%)	
MAO-B inhibitors (Selegiline/Rasagiline)	24 (42.1%)	13 (34.2%)	
COMT inhibitors (Entacapone)	3 (5.3%)	9 (23.7%)	
Amantadine	2 (3.5%)	7 (18.4%)	
Falling frequency			
Once/per year	--	23 (60.5%)	
Once/per month	--	11 (28.9%)	
Once/per week	--	1 (2.6%)	
≥Once/per day	--	3 (7.9%)	
Falling severity			
Mild	--	24 (63.2%)	
Moderate	--	13 (34.2%)	
Severe	--	1 (2.6%)	
Falling situations			
Sitting/standing	--	11 (26.3%)	
Walking	--	25 (65.8%)	
Turning	--	3 (7.9%)	

Φ = All the patients took more than one kind of anti-Parkinsonian medication. Abbreviations: UPDRS = Unified Parkinson’s Disease Rating Scale; ADL = activity of daily living; LEDD = Levodopa equivalent daily dose, area (cm^2^); PIGD = postural instability and gait difficulty; MAO-B = monoamine oxidase B; COMT = catechol-o-methyl-transferase; -- = none.

**Table 2 jcm-12-00127-t002:** The parameters of standing postural sway, UPDRS, and Tinetti total score between patients with and without dyskinesia before and after dopaminergic therapy.

	PD			Healthy Control (*n* = 23)	*p*-Value ^Ω^
	With Dyskinesia ^α^ (*n* = 20)	Without Dyskinesia ^α^ (*n* = 75)	*p*-Value ^β,κ^		
Age, years	65.4 ± 7.8	68.9 ± 10.4	0.16	67.2 ± 9.7	0.11
Disease duration, years	9.7 ± 4.7	4.3 ± 3.7	<0.0001	-	
LEDD (mg)	1477.1 ± 505.87	632.4 ± 443.2	<0.0001	-	
Parameters of postural sway					
Area (cm^2^)					
Off-state	2.4 ± 1.5 *	2.0 ± 1.2 *	0.33 ‖	1.2 ± 0.8 *	0.007 ^†^
On-state	5.0 ± 3.3	3.0 ± 2.0	0.001		
Velocity (cm/s)					
Off-state	1.2 ± 0.4 *	1.2 ± 0.9	0.9	0.8 ± 0.2	0.049 ^†^
On-state	1.7 ± 0.7	1.2 ± 0.6	0.004		
Length (cm)					
Off-state	36.7 ± 27.4 *	36.2 ± 11.8	0.94	23.5 ± 5.0	0.048 ^†^
On-state	50.3 ± 20.6	35.9 ± 17.6	0.002		
Length y (cm)					
Off-state	2.5 ± 0.8 *	2.4 ± 0.6 *	0.44 ‖	2.1 ± 0.7	0.44
On-state	3.4 ± 1.0	2.7 ± 0.9	0.01		
Length x (cm)					
Off-state	1.8 ± 0.7 *	1.7 ± 0.7 *	0.49 ‖	1.2 ± 0.3	0.001 ^†^
On-state	2.9 ± 1.3	2.0 ± 0.9	0.01		
Clinical scores					
PIGD score					
Off-state	7.5 ± 2.4 *	4.1 ± 3.9 *	<0.0001 ‖		
On-state	4.1 ± 2.9	2.6 ± 2.5	0.03		
UPDRS III					
Off-state	20.3 ± 8.1 *	17.3 ± 10.0 *	0.22		
On-state	10.9 ± 6.8	11.5 ± 7.4	0.74		
UPDRS total score					
Off-state	48.1 ± 12.3 *	36.4 ± 19.9 *	0.02 ‖		
On-state	29.4 ± 11.0	21.2 ± 11.9	0.74		
Falls risk score					
Tinetti balance score					
Off-state	7.7 ± 2.5 *	8.7 ± 3.2 *	0.20		
On-state	11.1 ± 1.4	10.4 ± 2.0	0.15		
Tinetti gait score					
Off-state	13.0 ± 2.5 *	13.7 ± 3.3 *	0.36		
On-state	15.1 ± 1.6	15.2 ± 1.7	0.79		
Tinetti balance and gait score					
Off-state	20.7 ± 4.0 *	22.5 ± 6.1 *	0.24		
On-state	26.2 ± 2.6	25.6 ± 3.5	0.48		

Abbreviations: Area (cm^2^), Length (cm), and Velocity (cm/s) traveled by the excursion (mm) of the center of pressure (COP); Length y indicates path length in the anterior-posterior direction; Length x indicates path length in medio–lateral direction; UPDRS = Unified Parkinson’s Disease Rating Scale; LEDD = Levodopa equivalent dose; PIGD = postural instability, and gait difficulty; α = baseline data before and after dopaminergic therapy were compared using paired t test, * = indicates *p* < 0.05; β = baseline data between patients with and with dyskinesia during off-state or on-state were compared using independent *t* test; Ω = baseline data among three groups during off-state or on-state were compared by mean of one-way ANOVA, ^†^ = indicates *p* < 0.05; κ = data between patients with and without dyskinesia before and after dopaminergic therapy were compared by mean of repeated measure ANOVA. ‖ = indicates *p* < 0.05.

**Table 3 jcm-12-00127-t003:** Risk factors of clinical scores and posture sway parameters associated with prospective falls in Cox’s proportional hazards model.

Univariate	Prospective Falls (*n* = 24)	No Prospective Falls (*n* = 71)	Univariate Analysis		Multivariate Analysis	
			HR (95% CI)	*p*-Value	HR (95% CI)	*p*-Value
Age, years	68.8 ± 10.3	67.9 ± 9.7	1.01 (0.97–1.06)	0.56		
Disease duration (years)	8.4 ± 5.4	4.4 ± 3.6	1.13 (1.04–1.22)	0.001 *		
LEDD (mg/day)	1377.1 ± 604.8	615.1 ± 410.8	1.0 (1.0–1.01)	<0.0001 *	1.0 (1.0–1.01)	0.008 *
Previous falls history	24	14	130.0 (3.8–4443.4)	0.007 *	139.4 (3.6–5392.0)	<0.0001 *
Height (m)	1.59 ± 0.08	1.59 ± 0.09	0.53 (0.003–101.15)	0.82		
Body weight (kg)	63.0 ± 11.9	63.8 ± 11.2	0.99 (0.96–1.02)	0.52		
Body mass index	24.9 ± 4.1	25.2 ± 4.2	0.97 (0.89–1.07)	0.53		
Presence of dyskinesia	11 (45.8%)	9 (12.7%)	2.87 (1.27–6.5)	0.01		
Orthostatic hypotension	5	1	1.18 (0.168.79)	0.88		
Fear of fall	20	19	10.7 (3.5931.9)	<0.0001		
Freezing of gait	18	18	5.01 (2.0112.86)	0.001		
Clinical scores						
UPDRS total score						
Off-state	50.6 ± 16.1	34.6 ± 18.2	1.03 (1.02–1.05)	<0.0001 *		
On-state	31.4 ± 12.1	20.0 ± 10.5	1.07 (1.04–1.1)	<0.0001 *		
Tinetti balance and gait score						
Off-state	17.7 ± 4.7	23.7 ± 5.3	0.83 (0.76–0.90)	<0.0001 *		
On-state	23.6 ± 3.5	26.4 ± 2.9	0.82 (0.75–0.89)	<0.0001 *	0.82 (0.74–0.91)	0.04 *
Cognitive Abilities Screening Instrument	83.1 ± 11.1	82.6 ± 14.3	1.0 (0.97–1.04)	0.81		
Standing postural sway, off-state						
Area (cm^2^)	2.3 ± 1.5	2.0 ± 1.2	1.23 (0.92–1.63)	0.16		
Velocity (cm/s)	1.4 ± 0.8	1.1 ± 0.8	1.43 (0.99–2.03)	0.051		
Length (cm)	41.7 ± 24.1	34.4 ± 24.8	1.01 (1.0–1.02)	0.053		
Length y (cm)	2.6 ± 0.8	2.4 ± 0.6	1.65 (0.91–3.01)	0.1		
Length x (cm)	1.9 ± 0.8	1.6 ± 0.6	1.62 (0–2.91)	0.11		
Standing postural sway, on-state						
Area (cm^2^)	3.7 ± 2.5	3.4 ± 2.4	1.05 (0.91–1.21)	0.54		
Velocity (cm/s)	1.6 ± 0.8	1.2 ± 0.5	1.72 (1.1–2.68)	0.016 *		
Length (cm)	47.8 ± 27.1	35.5 ± 15.7	1.02 (1.0–1.03)	0.011 *		
Length y (cm)	3.0 ± 0.9	2.8 ± 1.0	1.1 (0.78–1.62)	0.54		
Length x (cm)	2.4 ± 1.0	2.1 ± 1.0	1.16 (0.82–1.64)	0.4		

Abbreviations: CASI = Cognitive Abilities Screening Instrument; UPDRS = Unified Parkinson’s Disease Rating Scale; LEDD = Levodopa equivalent dose, area (cm^2^), length (cm), and velocity (cm/s) traveled by the excursion (mm) of the center of pressure (COP). Length y indicates path length in the anterior–posterior direction. Length x indicates path length in the medio–lateral direction, * Indicate *p* < 0.05.

**Table 4 jcm-12-00127-t004:** Partial correlation analysis among parameters of standing postural sway, LEDD and UPDRS total score, and Tinetti balance and gait score after controlling with the moment of inertia.

Spearman Correlation	LEDD		UPDRS Total Score, on Stage		Tinetti Balance and Gait Score	
	r	*p*-Value	r	*p*-Value	r	*p*-Value
Baseline characteristics						
Disease duration (years)	0.62	<0.0001	0.33	0.001	−0.09	0.42
UPDRS total score, on-state	0.48	<0.0001	--	--	−0.54	<0.0001
LEDD (mg/day)	--	--	0.48	<0.0001	−0.25	0.02
Standing postural sway, on-state						
Length (cm)	0.39	<0.0001	0.42	<0.0001	−0.38	<0.0001
Velocity (cm/s)	0.38	<0.0001	0.41	<0.0001	−0.38	<0.0001

Abbreviations: LEDD, levodopa equivalent dose; area (cm^2^), length, and velocity traveled by the excursion (mm) of the center of pressure (COP). Length y indicates path length in the anterior–posterior direction. Length x indicates path length in the medio–lateral direction.

**Table 5 jcm-12-00127-t005:** Diagnostic accuracy for predicting prospective falls, using receiver-operating characteristic curve analysis.

Significant Parameters ^†^	Cut-Off Value	Sensitivity (%)	Specificity (%)	AUC (95% CI)	*p*-Value
Combined LEDD, Tinetti balance and gait score, and length and velocity of postural sway	--	91.3%	80%	0.90 (0.83–0.98)	<0.0001
LEDD (mg)	895	82.6%	74.3%	0.85 (0.74–0.96)	<0.0001
Tinetti balance and gait score (on-state)	24.5	84.3%	60.0%	0.76 (0.64–0.89)	<0.0001
Length (cm) of postural sway, on-state	35.7	78.3%	62.9%	0.73 (0.62–0.84)	0.001
Velocity (cm/s) of postural sway, on-state	1.19	73.9%	65.7%	0.72 (0.61–0.84)	0.001

Abbreviations: AUC: area under curve; CI: confidence interval; LEDD, levodopa equivalent dose; UPDRS, Unified Parkinson’s Disease Rating Scale; † = DeLong method: LEDD vs. LEDD, Tinetti balance and gait score, and length and velocity of postural sway combined, *p* = 0.047; Tinetti balance and gait score vs. LEDD, Tinetti balance and gait score, and length and velocity of postural sway combined, *p* = 0.02; length of postural sway vs. LEDD, Tinetti balance, and gait score, and length and velocity of postural sway combined, *p* = 0.006; the velocity of postural sway vs. LEDD, Tinetti balance and gait score, and length and velocity of postural sway combined, *p* = 0.005.

## Data Availability

The datasets used and/or analyzed during the current study are available from the corresponding author upon reasonable request.

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
