# Peer review of "Feasibility of Combining Disease-Specific and Balance-Related Measures as Risk Predictors of Future Falls in Patients with Parkinson’s Disease"

_jcm, 2022, doi:10.3390/jcm12010127_

Round 1

Reviewer 1 Report

The authors present an interesting article. They aimed to analyze feasibility of combining disease-specific and balance-related measures as risk predictors for future falls in patients with Parkinson's disease. The manuscript consists of 16 pages, including 5 tables, 4 figures and 39 references.

The topic of the study is highly relevant in everyday clinical practice.

Comments to the manuscript:

- The introduction is informative and refers to previous studies considering postural control and balance in Parkinson's disease.

- The authors describe in "Patients and Methods" section that patients with Parkinson's disease according to Hoehn and Yahr classification stadium 1-3 and independent walking ability were included in the study. Patients with Hoehn and Yahr stadium 4 and above were excluded. According to Hoehn and Yahr classification patients in stadium 4 have severe disability, but they are still able to walk unassisted. Therefore, it is not clear why patients with Hoehn and Yahr stadium 4 were excluded. The authors should give an explanation at this point.

- Furthermore patients with cognitive disorder that precluded following study instructions were also excluded. Is cognitive impairment that leads to study exclusion underpinned by a clinical score? In the manuscript Cognitive Abilities Screening Instrument is mentioned. Were patients' study inclusion based on that score? If so, the instrument should be further described and cut-off values should be stated if applicable. To assess cognitive decline in patients with Parkinson's disease PANDA test (Parkinson Neuropsychometric Dementia Assessment) is useful. Did the authors also consider using PANDA test for their study?

- In sections "Fall assessment“, results and table 1 authors subdivided into mild, moderate and sever "falling severity". Authors referenced the following article at this point:

"Youn, J.; Okuma, Y.; Hwang, M.; Kim, D.; Cho, J.W. Falling Direction can Predict the Mechanism of Recurrent Falls in Advanced Parkinson's Disease. Sci Rep 2017, 7, 3921, doi:10.1038/s41598-017-04302-7."

Youn et al. defined "falling severity" as follows in their study: "The severity was defined as mild (did not need any treatment), moderate (needed simple treatment) and severe (needed to visit the hospital)."

Did you adopt "falling severity" exactly according to criteria of Youn et al.? Please give a detailed explanation how "falling severity" is defined in your manuscript.

- Results are clearly presented and accentuated in 5 tables and 2 figures.

- Discussion section is well structured. Authors also comment strengths and limitations of their study. 

- Discussion could be adapted with suggestions for future research.

Author Response

Comments and Suggestions for Authors

The authors present an interesting article. They aimed to analyze feasibility of combining disease-specific and balance-related measures as risk predictors for future falls in patients with Parkinson's disease. The manuscript consists of 16 pages, including 5 tables, 4 figures, and 39 references. The topic of the study is highly relevant in everyday clinical practice. Comments to the manuscript:

  1. The introduction is informative and refers to previous studies considering postural control and balance in Parkinson's disease.

Answers: Thanks for your comment

  1. The authors describe in "Patients and Methods" section that patients with Parkinson's disease according to Hoehn and Yahr classification stadium 1-3 and independent walking ability were included in the study. Patients with Hoehn and Yahr stadium 4 and above were excluded. According to Hoehn and Yahr classification patients in stadium 4 have severe disabilities, but they are still able to walk unassisted. Therefore, it is not clear why patients with Hoehn and Yahr stadium 4 were excluded. The authors should give an explanation at this point.

Answers: Thanks for your comment. Patients with Hoehn and Yahr stage 4 had severely disabling diseases but were still able to walk or stand unassisted. However, Patients with Hoehn and Yahr stage 4 have in danger during the gait study, especially the off state of medication. To make it clear, we revised the phrase “Patients with advanced PD stage (Hoehn and Yahr staging ≥ 4)” into “Patients with severe disability or wheelchair unless aided (Hoehn and Yahr staging ≥ 4)”.

  1. Furthermore, patients with a cognitive disorder that precluded following study instructions were also excluded. Is cognitive impairment that leads to study exclusion underpinned by a clinical score? In the manuscript, Cognitive Abilities Screening Instrument is mentioned. Were patients' study inclusion based on that score? If so, the instrument should be further described and cut-off values should be stated if applicable. To assess cognitive decline in patients with Parkinson's disease PANDA test (Parkinson Neuropsychometric Dementia Assessment) is useful. Did the authors also consider using the PANDA test for their study?

Answers: Thanks for your comment. All our patients received the Mini-Mental State Examination (MMSE), Clinical Dementia Rating (CDR), and The Cognitive Abilities Screening Instrument (CASI C-2.0). The Cognitive Abilities Screening Instrument (CASI C-2.0) consists of 20 items divided into 9 domains including attention, concentration, short- and long-term memory, orientation, language abilities, visual construction, abstraction, judgment, and category fluency25,26. The sum of the scores ranges from 0 to 100, with higher scores indicating better cognitive ability. We did not Parkinson's Neuropsychometric Dementia Assessment in our study. Perhaps, we can follow up on your suggestion shortly in our study. In our study, we exclude those patients who had a Clinical Dementia Rating of more than or equal to 1. To make it clear, we revise the phrase “cognitive decline that precluded following our instructions” into “cognitive decline that precluded following our instructions (Clinical Dementia Rating of more than or equal to 1)”.

  1. In sections "Fall assessment “, results, and table 1 authors subdivided into mild, moderate, and severe "falling severity". Authors referenced the following article at this point: "Youn, J.; Okuma, Y.; Hwang, M.; Kim, D.; Cho, J.W. Falling Direction can Predict the Mechanism of Recurrent Falls in Advanced Parkinson's Disease. Sci Rep 2017, 7, 3921, doi:10.1038/s41598-017-04302-7." Youn et al. defined "falling severity" as follows in their study: "The severity was defined as mild (did not need any treatment), moderate (needed simple treatment) and severe (needed to visit the hospital)." Did you adopt "falling severity" exactly according to criteria of Youn et al.? Please give a detailed explanation how "falling severity" is defined in your manuscript.

Answers: Thanks for your comment. In order to define the falling severity, we adopt "falling severity" exactly according to criteria of Youn et al in our study. We intend to grade the frequency and severity of falls as the baseline data of falls rather than the presence of falls or not.

  1. Results are clearly presented and accentuated in 5 tables and 2 figures.

Answers: Thanks for your comment.

  1. Discussion section is well structured. Authors also comment strengths and limitations of their study. 

Answers: Thanks for your comment.

  1. Discussion could be adapted with suggestions for future research.

Answers: Thanks for your comment. We added the following sentences in the conclusion section. They are as follows.

 Monitoring the changes in the COP motion during quiet standing, corresponding to quantitative postural sway measurement, to track disease progression during longitudinal follow-up can provide a scientific basis for dopaminergic therapy, balance training programs, and the selection of high-risk candidates for surgical treatment to reduce future fall risk

Reviewer 2 Report

The study of Tsai and colleagues aims to assess the perspective of future falls in PD patients that underwent through L-dopa. The paper is well written, the data analysis was carried properly and it highlighted interesting points. However, some delicate points should be revised regarding the postorugraphic descriptors used in the study. In the following, major points are highlighted.

Major Concerns

1) The posturographic analysis carried by the Authors was well conducted, however they considered quantities that could be related more with the mechanics of the stance rather than to the changes in the neural control mechanisms. For this reason I suggest the authors to review the two following papers

[1]"Center of pressure plausibility for the double-link human stance model under the intermittent control paradigm." Journal of Biomechanics 128 (2021): 110725.

[2]"Universal and individual characteristics of postural sway during quiet standing in healthy young adults." Physiological reports 3.3 (2015): e12329.

2) Based on the previous comment, Authors should report the correlation between the momentum of inertia (simply estimated as in [2]) and the posturographic descriptors computed in their analysis.

3)Authors should enrich the posturographc analysis with descriptors that can be considered universal, thus related with effective changes in motor control strategies of the patients rather than to mechanics of the stance.

Author Response

The study of Tsai and colleagues aims to assess the perspective of future falls in PD patients that underwent through L-dopa. The paper is well written, the data analysis was carried properly and it highlighted interesting points. However, some delicate points should be revised regarding the postorugraphic descriptors used in the study. In the following, major points are highlighted.

Major Concerns

1) The posturographic analysis carried by the Authors was well conducted, however they considered quantities that could be related more with the mechanics of the stance rather than to the changes in the neural control mechanisms. For this reason, I suggest the authors to review the two following papers
[1]"Center of pressure plausibility for the double-link human stance model under the intermittent control paradigm." Journal of Biomechanics 128 (2021): 110725.
[2]"Universal and individual characteristics of postural sway during quiet standing in healthy young adults." Physiological reports 3.3 (2015): e12329.
Answers: Thanks for your comment. We review and cite the two papers in our manuscript by your comment.

2) Based on the previous comment, Authors should report the correlation between the momentum of inertia (simply estimated as in [2]) and the posturographic descriptors computed in their analysis.

Answers: Thanks for your comment. We added the following sentences in Method and Result sections. They are as follows:

2.4. Assessment of the moment of inertia

The moment of inertia (I) is the name given to rotational inertia, the rotational analog of mass for linear motion. It appears in the relationships for the dynamics of rotational motion. The values of the inertia (I) for each subject were calculated by mh2 where m and h were the mass (body weight) and height of the subject. It was standardized as a z-score in each subject, respectively, prior to the correlation analysis. As it is well-known that the angular momentum (L) of a body is always conserved, the moment of inertia (I) and angular velocity (ω) are inversely proportional to each other. (L= I x ω).

The correlation analyses were used to test the influence of moment of inertia on the parameters of standing postural sway during the off and on state of medication. Since the moment of inertia could affect sway velocity, partial correlation analysis was used to determine the relationship between the length and velocity of postural sway and LEDD, UPDRS total scores, and Tinetti balance and gait scores after controlling with the moment of inertia.

The correlation analyses were used to test the influence of moment of inertia on the parameters of standing postural sway during the off and on state of medication, and the statistical analysis (r, p-value) was as follows: Area (cm2) (r=-0.05, p=0.62), Velocity (cm/s) (r=-0.10, p=0.34), Length (cm) (r=-0.10, p=0.34), Length y (cm) (r=-0.05, p=0.65) and Length x (cm) (r=-0.09, p=0.39) of standing postural sway during off state of medication, and Area (cm2) (r=-0.07, p=0.5), Velocity (cm/s) (r=-0.16, p=0.11), Length (cm) (r=-0.16, p=0.13), Length y (cm) (r=-0.06, p=0.54)and Length x (cm) (r=-0.16, p=0.13) of standing postural sway during on state of medication. Further, the partial correlation analyses used to test the influence of the parameters of standing postural sway during the on state and LEDD, UPDRS total score, Tinetti balance, and gait score after controlling with the moment of inertia are listed in Table 4.

3). Authors should enrich the posturographc analysis with descriptors that can be considered universal, thus related with effective changes in motor control strategies of the patients rather than to mechanics of the stance.

Answers: Thanks for your comment. We added your comment in Introduction and Discussion section. They are as follows:

The human balance can be maintained by the mechanics of the body's effects as well as coexists with the processes in the central nervous system engaged to solve. It implies that the neural control mechanisms corresponding to the mechanics of the stance can generate control commands to stabilize the body, eventually preventing falls.

Fourth, the time course of the COP motion during quiet standing, corresponding to body sway, is of critical importance for understanding neural mechanisms of postural control [23]. Posturographic analysis with descriptors that can be considered universal, thus related to effective changes in motor control strategies of the patients rather than to mechanics of the stance.

Round 2

Reviewer 1 Report

The authors of the manuscript "Feasibility of combining disease-specific and balance-related measures as risk predictors of future falls in patients with Parkinson's disease" addressed my points of criticism. I have no further questions.

Reviewer 2 Report

Authors improved the analysis and addressed the concerns.